# EPHA5 regulates antifungal innate immunity by phosphorylating EPHB2 and Dectin-1

**Ru Gao[1], Heping Wang[1], Zhihui Cui[1], Yanyun Du[2,3,4,5,6], Ruirui He[2,3,4,5,6], Lingyun Feng[2,3,4,5], Bo Zeng[2,3,4,5], Yangyang Li[2,3,4,5], Guoling Huang[5], Ting Pan[5], Yuan Wang[5], Ming Yi[2,3,4,5]\*, Chenhui Wang [2,3,4,5,6]\***

1 Key Laboratory of Molecular Biophysics of the Ministry of Education, National Engineering Research Center for Nanomedicine, College of Life Science and Technology, Huazhong University of Science and Technology, Wuhan, China, 2 The Key Laboratory for Human Disease Gene Study of Sichuan Province and the Department of Laboratory Medicine, Sichuan Provincial People's Hospital, School of Medicine, University of Electronic Science and Technology of China, Chengdu, China, 3 Research Unit for Blindness Prevention of the Chinese Academy of Medical Sciences, Sichuan Academy of Medical Sciences and Sichuan Provincial People's Hospital, Chengdu, Sichuan, China, 4 Sichuan Medical Laboratory Clinical Medical Research Center, Sichuan Provincial People's Hospital, Chengdu, China, 5 School of Medicine, University of Electronic Science and Technology of China, Chengdu, Sichuan, China, 6 Key Laboratory of Personalized Medicine Research and Treatment of Sichuan Province, Sichuan Provincial People's Hospital, Chengdu, China

\* yimingemail1234@uestc.edu.cn (MY); wangch@uestc.edu.cn (CW)

## Abstract

Invasive fungal infections (IFIs) have emerged as a significant health threat and cause approximately 3.75 million deaths per year globally. Understanding the detailed mechanism of the immune response to eliminate invasive fungal pathogens may help to provide new insights for the development of antifungal methods and drugs. Previously, we reported that the tyrosine kinase receptor EPH receptor B2 (EPHB2) is a coreceptor of β-glucan and phosphorylates Syk to activate the antifungal downstream signaling pathway. However, how EPHB2 is activated after fungal infection is still unknown. In this study, we show that EPHA5 plays a critical role in antifungal immunity by phosphorylating EPHB2 and Dectin-1 after fungal infection, which facilitates the recruitment and activation of Syk and subsequent activation of downstream antifungal signaling pathways. Additionally, we showed that *EphA5*-deficient mice exhibited increased susceptibility to *Candida albicans* infection, with increased fungal burdens and impaired immune cell recruitment. This study provides not only a mechanistic explanation for EPHB2 and Dectin-1 activation after fungal infection but also new insights into potential therapeutic strategies for treating IFIs.

## Author summary

Invasive fungal infections (IFIs) have emerged as a significant health threat, and knowledge of the immune response to eliminate invasive fungal pathogens may

**Data availability statement:** All relevant data are within the manuscript and its Supporting Information files.

**Funding:** This investigation was supported by grants from the Youth Fund of the National Natural Science Foundation of China (82301987 to B.Z.); the National Science Fund for Distinguished Young Scholars (82225029, to C.H.W.); the Key Project of the National Natural Science Foundation of China (82430076, to C.H.W.); the Youth Fund of the National Natural Science Foundation of China (82302628, to Y.Y.D., 82301989, to R.R.H. and 82402704 to Y.Y.L.); the Postdoctoral Foundation of China (2022M720658, to Y.Y.D., and 2022M720659 to R.R.H.); the Sichuan Postdoctoral Innovation Plan (BX202202, to Y.Y.D.); the Postdoctoral Foundation of Sichuan Provincial People's Hospital (2022BH01, to R.R.H. and 2022BH07, to M.Y.); and the Postdoctoral Foundation of Sichuan Province (TB2022086, to R.R.H., TB2023092, to L.Y.F.). The funders played no role in the study design, data collection and analysis, decision to publish, or preparation of the manuscript.

**Competing interests:** The authors have declared that no competing interests exist.

provide new targets and methods for the prevention or treatment of IFIs. Previously, we reported that EPHB2 is a coreceptor of β-glucan and plays a critical role in innate antifungal immunity, but the detailed mechanism by which EPHB2 is activated after fungal infection is still unknown. Here, by using cellular and mouse fungal infection models, we report that EPHA5 is an upstream kinase for EPHB2 and Dectin-1 after fungal infection, which facilitates the subsequent activation of downstream antifungal signaling pathways. This study provides not only a detailed mechanism upstream of EPHB2 and Dectin-1 but also a potential therapeutic target for IFIs.

## Introduction

Invasive fungal infections (IFIs) have emerged as a significant global health threat, particularly among immunocompromised individuals such as organ transplant recipients, patients with hematological malignancies, and those living with HIV/AIDS [1]. These populations face an increased risk of developing IFIs due to impaired immune function, resulting in increased incidence and mortality rates. Recent estimates suggest that fungal infections are responsible for approximately 3.75 million deaths annually [2], nearly double the previous global burden [3]. Among the causative pathogens, *Candida albicans* (*C. albicans*) remains the most prevalent opportunistic fungal species and a leading cause of hospital-acquired infections, with mortality rates reaching 40% [4]. In addition, non-*C. albicans* species such as *Candida glabrata* and *Candida auris* have emerged as clinically significant threats. *C. glabrata* is particularly challenging due to its intrinsic resistance to azole antifungal agents, whereas *C. auris* has gained global attention for its multidrug resistance and frequent involvement in healthcare-associated outbreaks [5,6]. Despite progress in antifungal drug development, the efficacy of current therapies is increasingly undermined by increasing resistance and prolonged antifungal exposure [7]. These challenges highlight the urgent need to elucidate the mechanisms of host antifungal immunity to facilitate the development of novel therapeutic strategies.

C-type lectin receptors (CLRs) are central to the recognition of fungal pathogens by the innate immune system [8,9]. These receptors, including Dectin-1, Dectin-2, Mincle, and the Mannose Receptor (MRC1), recognize distinct components of fungal cell walls, such as β-glucans, α-Mannans, and glycolipids, initiating antifungal immune responses [10]. Activation of the CLR signaling cascade typically involves the phosphorylation of Dectin-1 and the recruitment of splenic tyrosine kinase (Syk), two critical steps that lead to the activation of the NF-κB and MAPK pathways, ultimately triggering the production of proinflammatory cytokines [11], including tumor necrosis factor-α (TNF-α), interleukin-6 (IL-6), IL-1β, and CXCL1, among others. This process is vital for controlling fungal infections, but the precise regulatory mechanisms involved remain incompletely understood.

In recent years, several immune regulators, including STING [12], TRIM3 [13], SHP-2 [14], DOCK2 [15], MYO1F [16], TAGAP [17], and EPHB2 [18], have been

identified as critical players in antifungal immune responses. For example, STING has been shown to translocate to the phagosome membrane during fungal infections, where it forms a complex with the tyrosine kinase Src to inhibit Syk recruitment and activation, thereby modulating Dectin-1-mediated signaling [12]. Similarly, SHP-2 facilitates Syk recruitment to Dectin-1, activating downstream immune responses [14]. These findings suggest that various proteins, including SHP-2 and EPHB2, form complexes with CLR-related proteins to regulate antifungal signaling pathways. These complexes play crucial roles in maintaining the stability of Syk and regulating its activation level. Therefore, it would be interesting to ascertain whether other proteins are involved in the formation of the CLR-related complex. The Eph receptor family, the largest subclass of receptor tyrosine kinases (RTKs), plays diverse roles in biological processes such as embryonic development, tissue homeostasis, and oncogenesis [19]. EPHA5, a member of this family, has historically been known for its role in axon guidance during nervous system development [20]. Recent studies have suggested that mutations in the *EphA5* gene are associated with increased survival rates in patients with lung adenocarcinoma treated with immune checkpoint inhibitors [21]. Notably, our group identified EPHB2 as a coreceptor for Dectin-1 that phosphorylates Syk for downstream signal activation [18]. However, the mechanism of EPHB2 activation after fungal stimulation is still unknown.

In this study, we demonstrated that EPHA5 plays a critical role in antifungal immunity. Mechanistically, EPHA5 phosphorylates EPHB2 and Dectin-1, thus facilitating the recruitment and activation of Syk and subsequent activation of antifungal signaling pathways. Additionally, we showed that *EphA5*-deficient mice exhibited increased susceptibility to *C. albicans* infection, with increased fungal burdens and impaired immune cell recruitment. These findings highlight the essential role of EPHA5 in regulating host defense against fungal infection and may also provide new insights into potential therapeutic strategies for treating IFIs.

## Results

### EPHA5 is essential for antifungal signaling in human and mouse macrophages

Our previous work identified the Rho GTPase-activating protein TAGAP as a key regulator of antifungal immune signaling [17]. Using immunoprecipitation and mass spectrometry, we found that EPHA5 interacts with TAGAP, prompting further investigation into its role in antifungal responses. To evaluate the functional relevance of EPHA5 in C-type lectin receptor (CLR)-mediated signaling, we generated EPHA5 knockdown (EPHA5-g2 or EPHA5-g3) THP-1 cells via CRISPR-Cas9. Upon stimulation with CLR ligands, including Curdlan (a Dectin-1 ligand), α-Mannan (a Dectin-2/3 ligand), or heat-killed *C. albicans* (HKCA) (a Dectin-1 agonist), EPHA5 knockdown THP-1 cells displayed significantly reduced activation of the NF-κB and MAPK signaling pathways compared with those in THP1 WT cells (Fig 1A and 1B). Phosphorylation of Syk, a critical upstream kinase in CLR signaling, was also significantly diminished in EPHA5 knockdown THP-1 cells upon stimulation with HKCA or α-Mannan (Fig 1A and 1B).

Consistent findings were observed in mouse bone marrow–derived macrophages (BMDMs). Following stimulation with Curdlan, α-Mannan, or HKCA, NF-κB and MAPK activation was significantly impaired in *EphA5*-KO BMDMs compared with WT BMDMs (Fig 1C–1E). Furthermore, both *EphA5*-KO BMDMs (Fig 1F–1H) and peritoneal macrophages (S2A–S2C Fig) presented decreased production of key proinflammatory mediators, including TNF-α, IL-1β, IL-6, and CXCL1. ELISA further confirmed the reduced secretion of TNF-α and CXCL1 in the culture supernatants of *EphA5*-KO BMDMs (Fig 1I) and peritoneal macrophages (S2D and S2E Fig).

To assess whether *EphA5* deficiency broadly affects innate immune signaling pathways beyond the fungal-specific CLR axis, *EphA5*-KO and WT BMDMs were stimulated with TLR ligands, including LPS (TLR4) and PolyI:C (TLR3). No significant differences in NF-κB or MAPK activation were detected between the two groups (S1F and S1G Fig), suggesting that EPHA5 selectively regulates CLR-dependent antifungal signaling. We next examined whether EPHA5 is also involved in the response to other clinically relevant fungal pathogens. *Candida glabrata* and *Candida auris* were selected because of their increasing prevalence in bloodstream infections and notable antifungal resistance profiles. In particular, *C. auris* is a

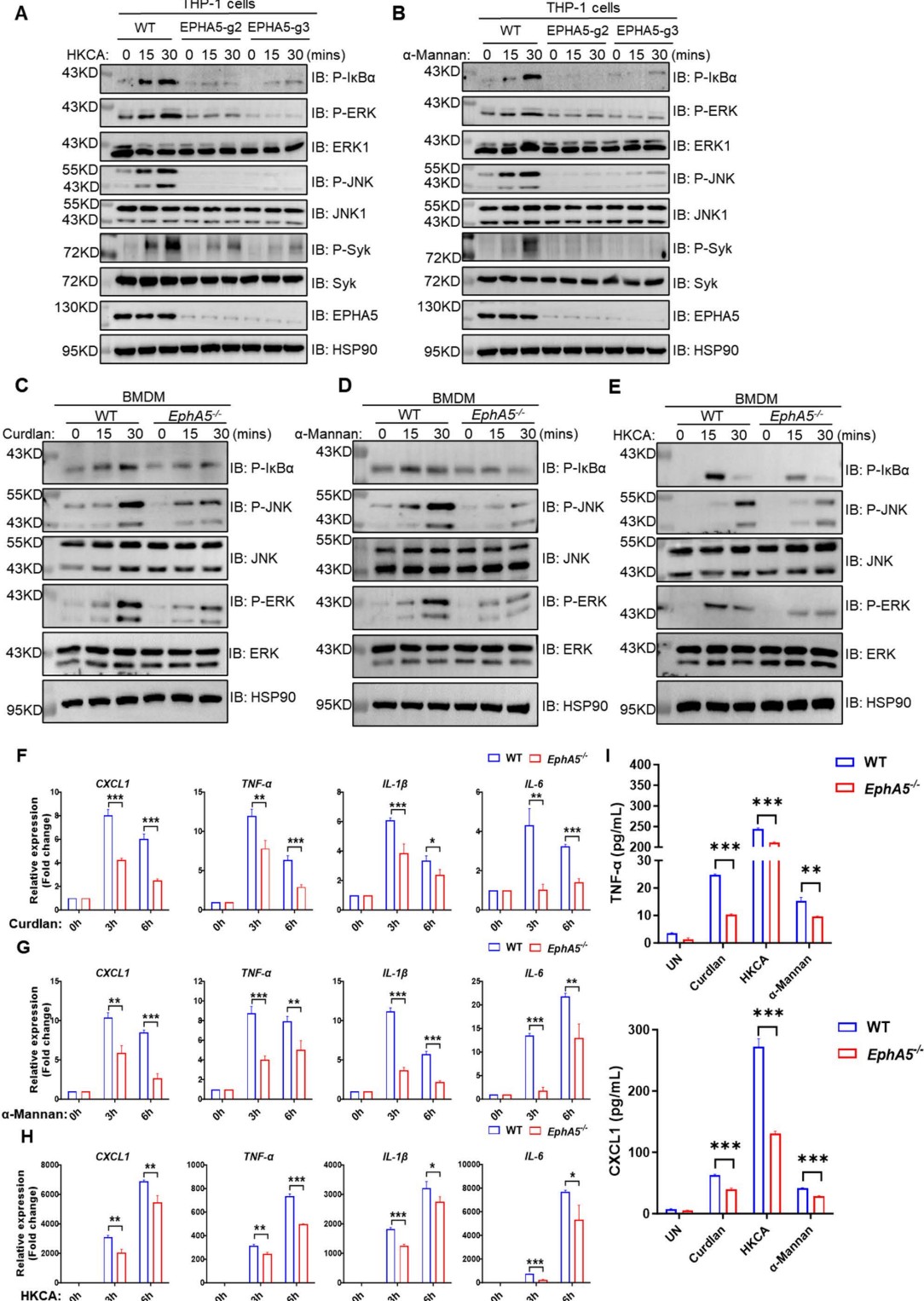

**Fig 1. EPHA5 is essential for activating antifungal signaling in macrophages.** (A-B) WT THP-1 cells and EPHA5 knockdown (EPHA5-g2 or EPHA5-g3) THP-1 cells were stimulated with heat-inactivated *C. albicans* (HKCA, E; T = 2) (A) or α-Mannan (100 μg/mL) (B) for the indicated times, followed by western blot analysis to assess the levels of the indicated proteins. (C-H) Bone marrow-derived macrophages (BMDMs) from WT or

*EphA5*-deficient (*EphA5*-KO) mice were treated with Curdlan (100 μg/mL) (C, F), α-Mannan (100 μg/mL) (D, G), or HKCA (E; T = 2) (E, H) for the indicated time, followed by quantitative reverse transcription polymerase chain reaction (RT–qPCR) to analyze the expression levels of the indicated genes, followed by western blot analysis to assess the levels of the indicated proteins. (I) Enzyme-linked immunosorbent assay (ELISA) analysis of TNF-α and CXCL1 levels in the supernatants of BMDMs from WT and *EphA5*-KO mice after stimulation with Curdlan (100 μg/mL), HKCA (E; T = 2), or α-Mannan (100 μg/mL) for 24 hours. The error bars represent the standard error of the mean (S.E.M.) of biological replicates. Statistical significance was determined by a two-tailed unpaired t test, with $*P < 0.05$; $**P < 0.01$; $***P < 0.001$; and $****P < 0.0001$. The results presented are representative of three independent experiments.

multidrug-resistant species implicated in healthcare-associated outbreaks and often exhibits resistance to first-line agents such as fluconazole [22]. BMDMs from WT or *EphA5*-KO mice were stimulated with *C. glabrata* or *C. auris*. Western blot analysis revealed substantially lower phosphorylation of Syk, IκBα, ERK, and JNK in *EphA5*-KO BMDMs than in WT BMDMs (Fig 2A and 2B). RT–qPCR analysis further demonstrated decreased transcription of proinflammatory cytokines in *EphA5*-KO BMDMs following stimulation with either *C. glabrata* (Fig 2C) or *C. auris* (Fig 2D). Together, these findings establish EPHA5 as a critical regulator of antifungal innate immunity that mediates CLR-dependent signaling and cytokine production in both human and mouse macrophages.

## EPHA5 is crucial for human antifungal immunity

To validate the role of EPHA5 in human antifungal responses, we used small interfering RNA (siRNA) to silence EPHA5 expression in human peripheral blood mononuclear cells (PBMCs). Efficient knockdown was confirmed by RT-qPCR (Fig 3A–3C). Upon stimulation with Curdlan, HKCA or α-Mannan, EPHA5-deficient PBMCs exhibited a significant reduction in proinflammatory cytokine production (Fig 3A–3C). Notably, EPHA5 knockdown also led to a marked decrease in the expression of IL-23A, a key cytokine essential for Th17 cell differentiation [23], following stimulation with HKCA or α-Mannan (Fig 3B and 3C). These findings indicate that EPHA5 not only regulates innate immune responses but also may contribute to shaping adaptive immunity by promoting Th17 polarization, highlighting its essential role in human antifungal defense.

## Kinase activity of EPHA5 is essential for antifungal immunity

Receptor tyrosine kinases (RTKs) are well-established mediators of cellular signaling, primarily through dimerization and autophosphorylation, which facilitate the recruitment and activation of downstream signaling molecules [24]. As a member of the RTK family, EPHA5 adheres to this canonical signaling mechanism [25]. To investigate whether EPHA5 undergoes phosphorylation in response to fungal stimuli, we reconstituted EPHA5-WT into EPHA5 knockdown (EPHA5-g2) THP-1 cells via lentiviral transduction. Immunoprecipitation assays demonstrated a time- and dose-dependent increase in EPHA5 tyrosine phosphorylation following stimulation with heat-killed Candida albicans (HKCA) or α-Mannan (Fig 4A and 4B). These findings suggest that EPHA5 may play a critical role in mediating antifungal immune responses through its kinase activity.

In silico sequence analysis identified a highly conserved lysine residue at position 708, situated within the ATP-binding pocket of EPHA5, as a putative critical site for maintaining its enzymatic function (UniProt, https://www.uniprot.org). To rigorously assess the functional significance of EPHA5 kinase activity in antifungal immunity, we generated a kinase-inactive mutant of EPHA5 by substituting lysine 708 with arginine (EPHA5-K708R) (Fig 4C). Next, EPHA5 knockdown (EPHA5-g2) THP-1 cells or RAW264.7 macrophages were transduced with constructs encoding either wild-type EPHA5 (EPHA5-WT), the kinase-dead mutant EPHA5-K708R, or an empty vector as a control. Upon stimulation with HKCA, EPHA5 knockdown (EPHA5-g2) THP-1 cells expressing EPHA5-WT, but not EPHA5-K708R, exhibited significantly increased activation of the NF-κB and MAPK signaling pathways (Fig 4D). Furthermore, in RAW264.7 cells, EPHA5-WT reconstitution markedly increased the production of key proinflammatory cytokines, including TNF-α, IL-6, and IL-1β, whereas EPHA5-K708R

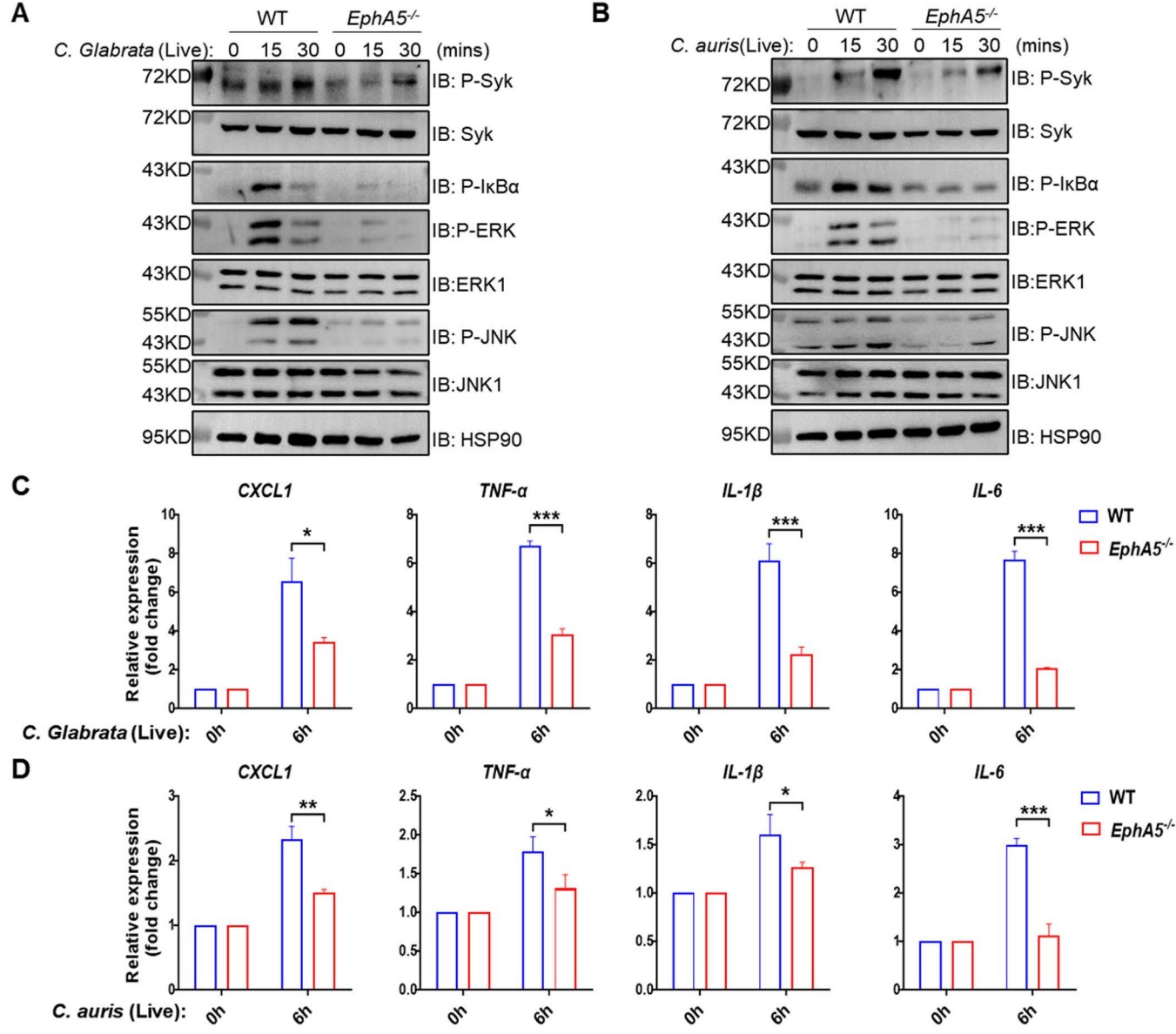

**Fig 2. *EphA5* is critical for live *C. glabrata*- or live *C. auris*-induced antifungal activity in BMDMs.** (A-B). BMDMs from WT and *EphA5⁻/⁻* mice were stimulated with live *C. glabrata* (MOI = 5) (A) or live *C. auris* (MOI = 5) (B) for the indicated times, followed by western blot analysis of the indicated proteins. (C-D) BMDMs from WT and *EphA5⁻/⁻* mice were stimulated with live *C. glabrata* (MOI = 5) (A) or live *C. auris* (MOI = 5) (B) for the indicated times, followed by RT–qPCR analysis of the gene expression of *CXCL1, TNF-α, IL-1β* and *IL-6*. Statistical significance was determined by one-way ANOVA (*$P < 0.05$; **$P < 0.01$; ***$P < 0.001$; ****$P < 0.0001$). The error bars represent the standard error of the mean (S.E.M.) from biological replicates. The data are representative of three independent experiments.

failed to elicit such responses (Fig 4E). Collectively, these findings underscore the critical role of EPHA5 kinase activity in orchestrating antifungal signaling pathways and promoting the production of proinflammatory cytokines during *C. albicans* infection.

Dimerization is a key regulatory step in the activity and signaling of Eph receptors. To further confirm whether EPHA5 dimerizes upon fungal stimulation, we co-transfected HEK293T cells with constructs encoding EPHA5-Flag and

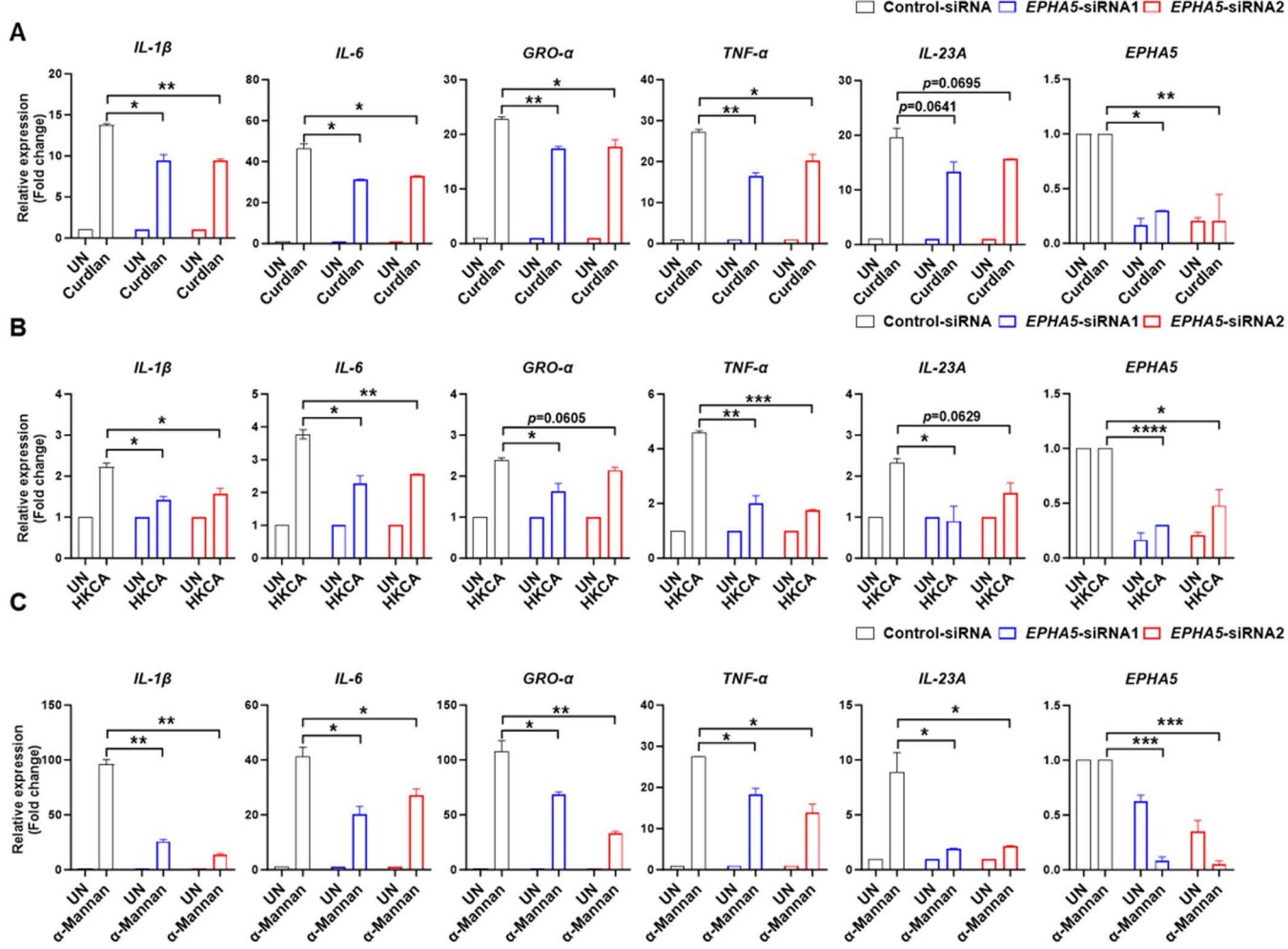

**Fig 3. EPHA5 plays a crucial role in human antifungal immunity.** Peripheral blood mononuclear cells (PBMCs) from healthy donors were transfected with either control siRNA or EPHA5-siRNA. At 5 days post-transfection, the cells were stimulated with Curdlan (100 µg/mL) (A), HKCA (E; T = 2) (B), or α-Mannan (100 µg/mL) (C) for 3 hours. The expression of the indicated genes was then analyzed via RT–qPCR. Statistical significance was determined by one-way ANOVA (*$P < 0.05$; **$P < 0.01$; ***$P < 0.001$; ****$P < 0.0001$). The error bars represent the standard error of the mean (S.E.M.) from biological replicates. The data are representative of three independent experiments.

EPHA5-HA, and found that EPHA5-HA interacted with EPHA5-Flag (Fig 4F and 4G), indicating that EPHA5 undergoes dimerization upon fungal stimulation. To further investigate whether EPHA5 undergoes dimerization in response to fungal stimuli, we reconstituted EPHA5-Flag into EPHA5 knockdown THP-1 cells and stimulated these cells with Curdlan or α-Mannan. Co-immunoprecipitation assays revealed that EPHA5 undergoes dimerization upon exposure to fungal stimuli (Fig 4H and 4I). These findings suggest that EPHA5 may activate downstream signaling pathways through its dimeric form, potentially serving as a critical regulatory mechanism in antifungal immune responses.

## EPHA5 interacts with and promotes the phosphorylation of EPHB2

Previous work from our laboratory identified EPHB2 as a key mediator of fungal signaling via the LPS receptor family, which has been shown to directly bind β-glucan, thereby facilitating antifungal immune responses [26,27]. In line with

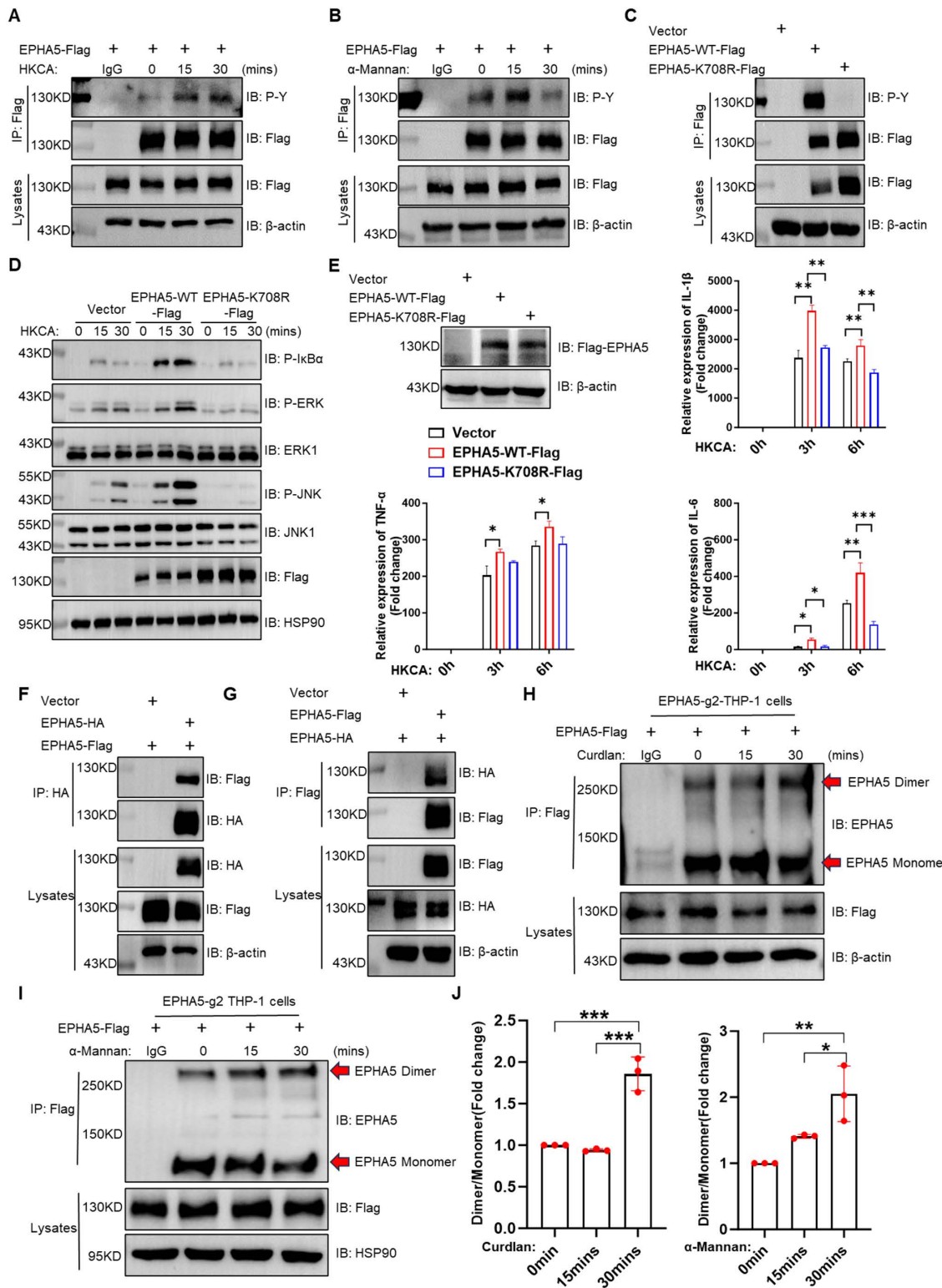

**Fig 4. EPHA5 regulates antifungal immune responses through its kinase activity.** (A-B) EPHA5 knockdown (EPHA5-g2) THP-1 cells were reconstituted with WT EPHA5 (EPHA5-WT), followed by stimulation with HKCA (E; T = 2) (A) or α-Mannan (100 μg/mL) (B) for the indicated durations. The cell lysates were immunoprecipitated with an anti-Flag antibody (IgG served as a control), followed by Western blot analysis of the indicated proteins. (C)

HEK293T cells were transfected with plasmids encoding either the control vector, EPHA5-WT-Flag, or the kinase-inactive mutant EPHA5-K708R-Flag. The cell lysates were immunoprecipitated with an anti-Flag antibody, followed by Western blot analysis of the indicated proteins. (D) EPHA5 knockdown (EPHA5-g2) THP-1 cells were reconstituted with EPHA5-WT or EPHA5-K708R, followed by stimulation with HKCA (E; T = 2). The cell lysates were subjected to Western blot analysis of the indicated proteins. (E) RAW264.7 cells were transduced with lentiviral vectors to overexpress either the control vector, EPHA5-WT, or EPHA5-K708R. The cells were stimulated with HKCA (E; T = 2) for the indicated times, followed by RT–qPCR analysis of the indicated gene transcripts (E). (F-G) HEK293T cells were transfected with plasmids encoding either the control vector, EPHA5-Flag, or EPHA5-HA. The cell lysates were immunoprecipitated with an anti-HA (F) or anti-Flag (G) antibody, followed by Western blot analysis of the indicated proteins. (H-J) EPHA5 knockdown (EPHA5-g2) THP-1 cells were reconstituted with EPHA5-WT-Flag. After stimulation with Curdlan (100 μg/ml) (H) or α-Mannan (100 μg/mL) (I) for the indicated times, the cells were immunoprecipitated with an anti-Flag antibody, followed by Western blot analysis of the indicated proteins. (J) Quantitative analysis of the grayscale intensity ratios of EPHA5 dimers to monomers was shown in H and I. Significance was determined via one-way ANOVA (*$P < 0.05$; **$P < 0.01$; ***$P < 0.001$; ****$P < 0.0001$). The error bars represent the standard error of the mean (S.E.M.) from biological replicates. The data are representative of three independent experiments.

these findings, our current study revealed that EPHA5 also plays a pivotal role in antifungal immunity. To investigate whether EPHA5 functions as a coreceptor for dectin ligands, we assessed its binding affinity for β-glucan (a dectin-1 ligand) and α-Mannan (a dectin-2/3 ligand). While the recombinant EPHB2 protein (EPHB2-Fc) bound to β-glucan, the recombinant EPHA5 proteins (EPHA5-Fc and EPHA5-His) did not directly bind to either β-glucan or α-Mannan (Fig 5A). These findings suggest that EPHA5 does not act as a coreceptor for dectin ligands in antifungal signaling.

EPHB2 has been implicated in antifungal signaling via the phosphorylation of Syk [18], yet the precise mechanisms underlying its activation remain incompletely understood. Given the involvement of Eph receptors in transphosphorylation events, we hypothesized that EPHA5 may modulate antifungal immunity by facilitating the phosphorylation of EPHB2. To test this hypothesis, we performed coimmunoprecipitation experiments in HEK293T cells overexpressing EPHB2, which revealed a physical interaction between EPHA5 and EPHB2 (Fig 5B). This interaction was further corroborated in EPHB2-knockout (KO) THP-1 cells reconstituted with EPHB2, where the EPHA5-EPHB2 interaction increased in a time-dependent manner following stimulation with Curdlan (Fig 5C) or α-Mannan (Fig 5D). Moreover, the reconstitution of WT EPHA5 (EPHA5-WT), but not the kinase-inactive mutant EPHA5-K708R, in EPHA5 knockdown THP-1 cells markedly increased EPHB2 phosphorylation upon stimulation with Curdlan (Fig 5E) and α-Mannan (Fig 5F). These findings indicate that EPHA5 interacts with EPHB2 and promotes its phosphorylation in response to fungal stimuli, suggesting a regulatory role of EPHA5 in EPHB2-mediated antifungal signaling.

### EPHA5-mediated Dectin-1 phosphorylation at tyrosine 15 promotes the recruitment of Syk

Upon fungal stimulation, phosphorylation of Dectin-1 at tyrosine 15 (Tyr15) is a crucial event that facilitates the recruitment and activation of Syk kinase, a key regulatory step in triggering downstream signaling pathways essential for an effective antifungal immune response [28]. While previous studies have implicated Src family kinases in mediating Dectin-1 phosphorylation [29,30], the regulatory mechanisms that control this process remain incompletely defined. Given that EPHA5 is a receptor tyrosine kinase, we hypothesized that it may also regulate Dectin-1 phosphorylation. To explore this, we examined the potential interaction between EPHA5 and Dectin-1. Immunoprecipitation assays revealed a physical interaction between EPHA5 and Dectin-1, suggesting that EPHA5 may modulate Dectin-1 signaling (Fig 6A). In addition, the immunoprecipitation results confirmed that EPHA5 also interacted with Dectin2 (S4D Fig). Notably, in HEK293T cells co-transfected with Dectin1-HA, EPHB2-HA, or EPHA5-Flag, EPHA5 formed a stable complex with both Dectin-1 and EPHB2, indicating a potential cooperative role for these proteins in the antifungal signaling cascade (Fig 6B).

To determine whether EPHA5 kinase activity is required for Dectin-1 phosphorylation, we performed immunoblotting to detect phosphorylated tyrosine residues. EPHA5-WT significantly increased Dectin-1 phosphorylation at Tyr15, whereas the kinase-inactive EPHA5 mutant (EPHA5-K708R) did not (Fig 6C). These findings indicate that the kinase activity of EPHA5 is indispensable for the phosphorylation of Dectin-1 at this critical residue.

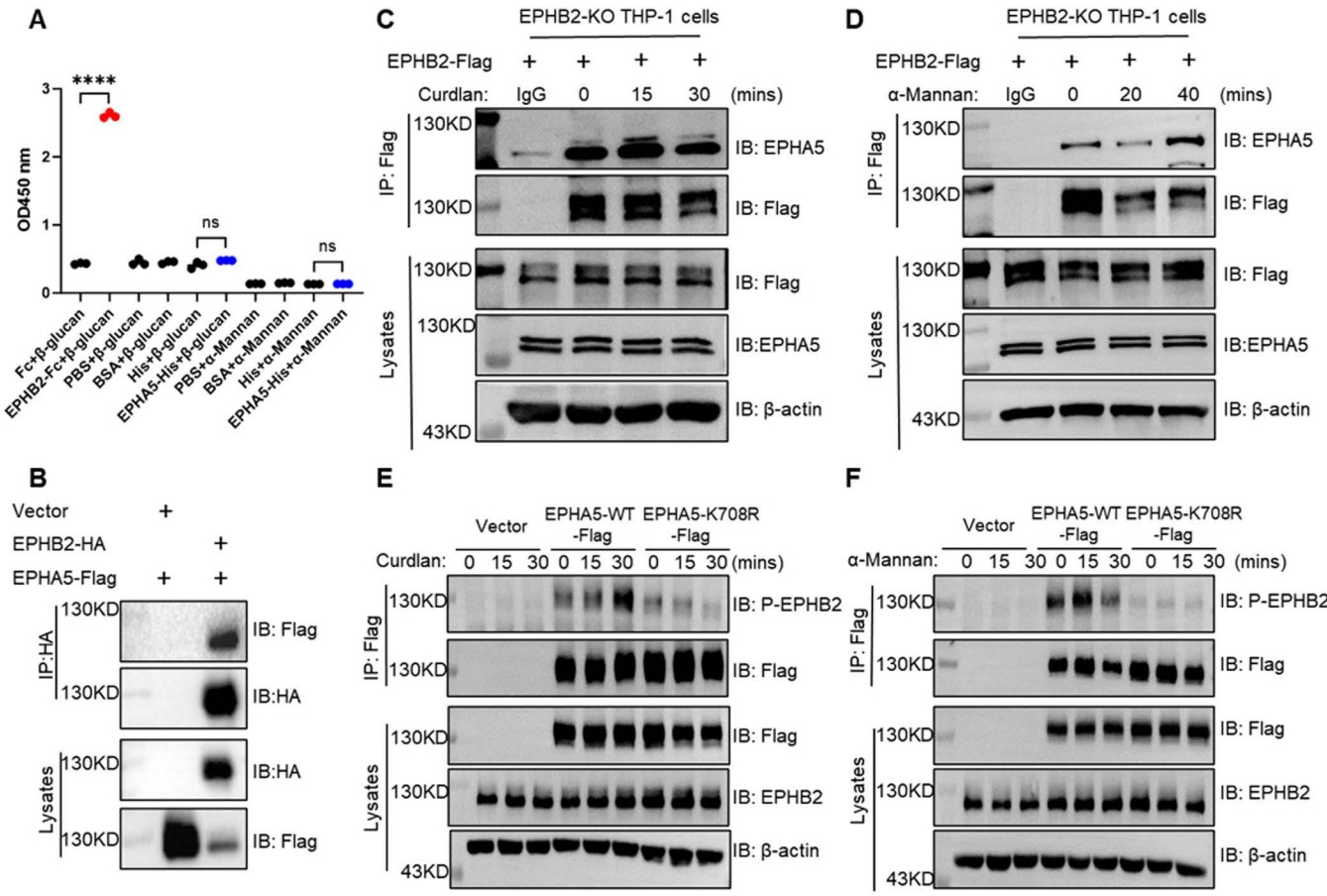

**Fig 5. EPHA5 binds to and phosphorylates EPHB2 upon stimulation with dectin ligands.** (A) ELISA was performed to assess the binding of recombinant EPHB2 and EPHA5 proteins to immobilized β-glucan and α-Mannan, with the optical density measured at 450 nm to quantify the interactions. (B) HEK293T cells were cotransfected with plasmids encoding EPHB2-HA and EPHA5-Flag. The cell lysates were immunoprecipitated with an anti-HA antibody, followed by Western blot analysis of the indicated proteins. (C-D) EPHB2-knockout (EPHB2-KO) THP-1 cells were reconstituted with WT EPHB2. These cells were stimulated with Curdlan (100 μg/mL) (C) or α-Mannan (100 μg/mL) (D) for the indicated times and were immunoprecipitated with an anti-Flag antibody, followed by Western blot analysis of the indicated proteins. (E-F) EPHA5 knockdown THP-1 cells were reconstituted with the control vector, EPHA5-WT, or the kinase-inactive mutant EPHA5-K708R. After stimulation with Curdlan (100 μg/mL) (E) or α-Mannan (100 μg/mL) (F) for the indicated times, the cells were immunoprecipitated with an anti-Flag antibody, followed by Western blot analysis of the indicated proteins. The data are representative of three independent experiments.

To further investigate the functional importance of Tyr15 phosphorylation in the ability of Dectin-1 to recruit Syk, we generated a Dectin-1 mutant in which Tyr15 was substituted with phenylalanine (Y15F). Our data demonstrated that while EPHA5 efficiently phosphorylated WT Dectin-1, the Dectin-1-Y15F mutant exhibited significantly reduced phosphorylation by EPHA5 (Fig 6D). Consistent with previous studies, the Dectin-1-Y15F mutant also markedly decreased Syk recruitment compared with that of WT Dectin-1, as demonstrated by immunoprecipitation assays (Fig 6E). These results emphasize the critical role of EPHA5-mediated Tyr15 phosphorylation in facilitating Syk recruitment, further underscoring its importance in antifungal immunity.

To elucidate the structural basis of the impact of Tyr15 phosphorylation on Syk recruitment, we utilized AlphaFold3 to predict the structural changes in the interaction between Syk and both the unphosphorylated (WT) and phosphorylated (PTR-15) forms of Dectin-1. In the WT Dectin-1 dimer, Tyr15 forms hydrogen bonds with Arg-195 and Asn-46 in the SH2

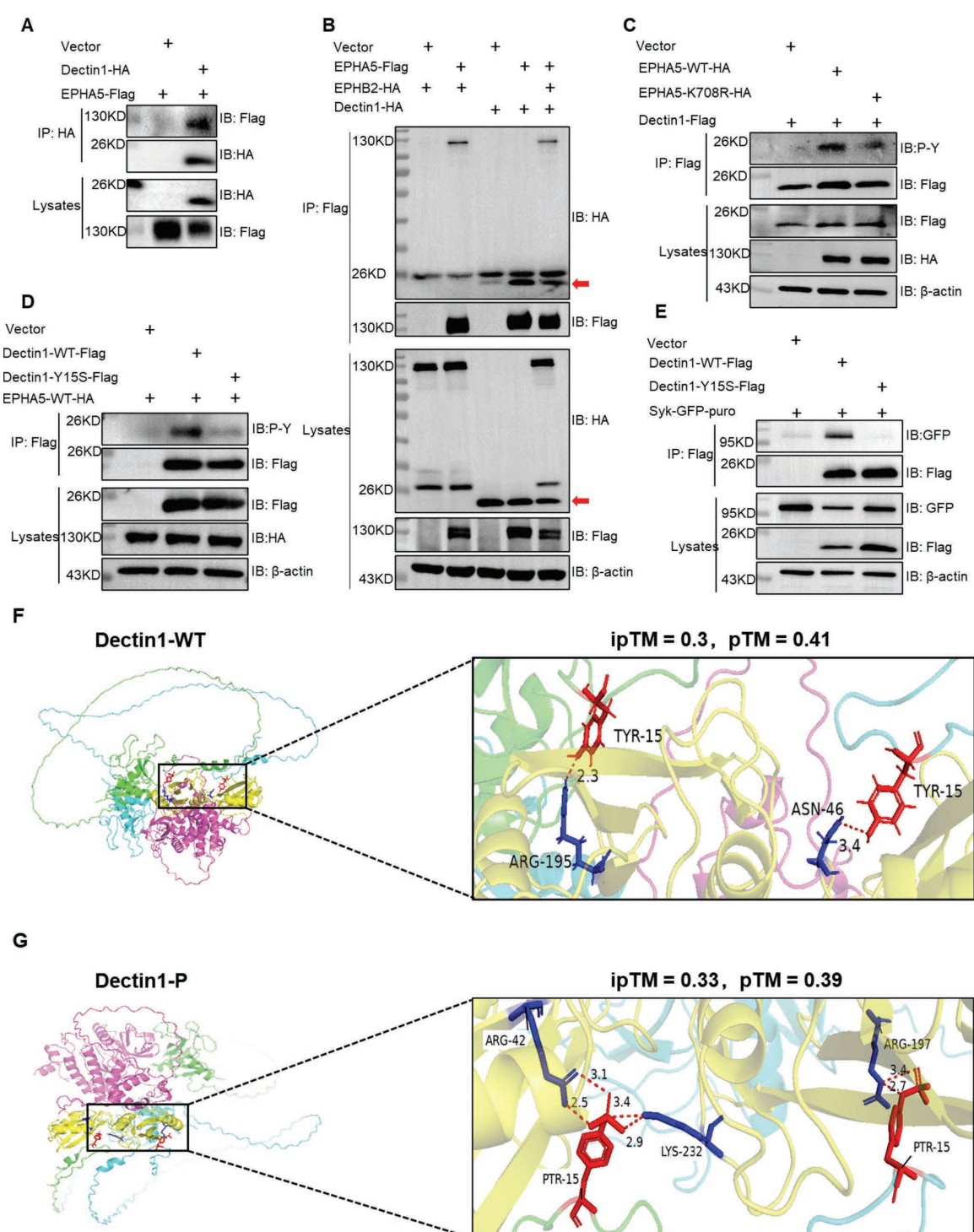

**Fig 6. EPHA5-mediated Dectin-1 phosphorylation at tyrosine 15 promotes Syk recruitment.** (A) HEK293T cells were transfected with EPHA5-Flag and Dectin1-HA and immunoprecipitated with an anti-HA antibody, followed by Western blotting for the indicated proteins. (B) HEK293T cells were cotransfected with EPHA5-Flag, EPHB2-HA, and Dectin1-HA, immunoprecipitated with an anti-Flag antibody and analyzed by Western blotting. (C) HEK293T cells were cotransfected with Dectin1-HA and either EPHA5-WT-Flag or EPHA5-K708R-Flag. Immunoprecipitation via an anti-Flag antibody was followed by Western blotting of the indicated proteins. (D) HEK293T cells cotransfected with Dectin1-HA and either EPHA5-WT-Flag or EPHA5-K708R-Flag were immunoprecipitated with an anti-Flag antibody, followed by Western blot analysis. (E) HEK293T cells were cotransfected with GFP-Syk, Dectin1-WT-Flag, or Dectin1-Y15S-Flag. Following immunoprecipitation with an anti-Flag antibody, the indicated proteins were detected via

PLOS Pathogens

Western blotting. (F) AlphaFold structural model of the Dectin1-WT dimer (green and blue) interacting with Syk (purple), with the SH2 domains shown in yellow. (G) AlphaFold structural model of the phosphorylated Dectin-1 dimer (green and blue) interacting with Syk (purple), with the SH2 domains shown in yellow. Red represents conventional hydrogen bonds. The data are from three independent experiments.

domain of Syk, with bond distances of 2.3 Å and 3.4 Å, respectively, stabilizing the Dectin-1-Syk interaction and promoting downstream signaling (Fig 6F). Upon Tyr15 phosphorylation (PTR-15), a distinct set of interactions occurs, with PTR-15 forming bonds with Arg-42, Lys-232, and Arg-197 in Syk's SH2 domain, with bond distances of 2.5 Å, 3.4 Å, and 3.1 Å, respectively (Fig 6G). This phosphorylation-induced shift in binding partners suggests a conformational change in the Dectin-1-Syk interaction, which may enhance the signaling cascade initiated by Dectin-1. These structural predictions support the conclusion that phosphorylation of Dectin-1 at Tyr15 is essential for efficient Syk recruitment, reinforcing the critical role of EPHA5 in modulating antifungal immune responses.

### *EphA5* deficiency increases susceptibility to fungal sepsis in vivo

To further investigate the role of EPHA5 in antifungal immunity in vivo, we intravenously infected WT and *EphA5*-KO mice with live *C. albicans*. Consistent with our in vitro findings, *EphA5*-KO mice exhibited significantly increased susceptibility to *C. albicans* infection, as evidenced by more rapid weight loss (Fig 7A) and a lower survival rate than WT controls did (Fig 7B). To determine whether the increased susceptibility of the *EphA5*-KO mice was due to underlying immunodeficiency, we examined the distributions of major immune cell populations in the peripheral blood, lymph nodes, and spleen under steady-state conditions. Flow cytometric analysis revealed no significant differences in the frequencies of key immune cell subsets, including T cells, B cells, macrophages, and neutrophils, between WT and *EphA5*-KO mice (S5A-S5C Fig). We next assessed the fungal burden in the kidneys of both WT and *EphA5*-KO mice at 48 hours post infection. Compared with WT mice, *EphA5*-KO mice presented significantly greater fungal burdens (Fig 7C). Furthermore, histopathological examination (Fig 7D) and periodic acid-Schiff (PAS) staining (Fig 7E) revealed increased renal inflammation and a greater fungal load in the kidneys of *EphA5*-KO mice than in those of WT mice, indicating that EPHA5 deficiency exacerbates fungal infection and tissue damage.

Macrophages and neutrophils are essential components of the innate immune response against fungal pathogens. They contribute to early host defense by producing cytokines and chemokines such as CXCL1 and GM-CSF, which facilitate the recruitment of additional neutrophils and macrophages to sites of infection [31]. The serum levels of CXCL1 were markedly lower in the *EphA5*-KO mice than in the WT mice (Fig 7F), indicating impaired chemokine-driven immune cell recruitment. To elucidate the cellular mechanisms underlying the increased susceptibility of *EphA5*-deficient mice to *Candida* infection, we established a chimeric bone marrow model. Lethally irradiated WT recipient mice were reconstituted with bone marrow from either WT (WT→WT group) or *EphA5⁻/⁻* (KO→KO group) donors, as well as from mixed bone marrow groups (WT→KO and KO→WT). K–M survival analysis revealed that, compared with those receiving WT bone marrow (WT→WT), mice reconstituted with EphA5⁻/⁻ bone marrow (KO→KO) presented significantly shorter survival times following *C. albicans* infection (Fig 7G). The WT→WT group and WT→KO group displayed similar survival outcome, and the KO→WT group and KO→KO group showed the comparable survival outcomes, which indicates that hematopoietic cell expressed EPHA5 plays an important role in the antifungal immune response (Fig 7G). In line with these findings, *EphA5*-KO mice presented significantly less infiltration of macrophages and monocytes in the kidneys following *C. albicans* infection than WT controls did, along with a clear trend toward decreased neutrophil recruitment (Figs 7H-J, S6A and S6B). These findings indicate that EPHA5 plays a critical role in coordinating innate immune cell infiltration during systemic fungal infection.

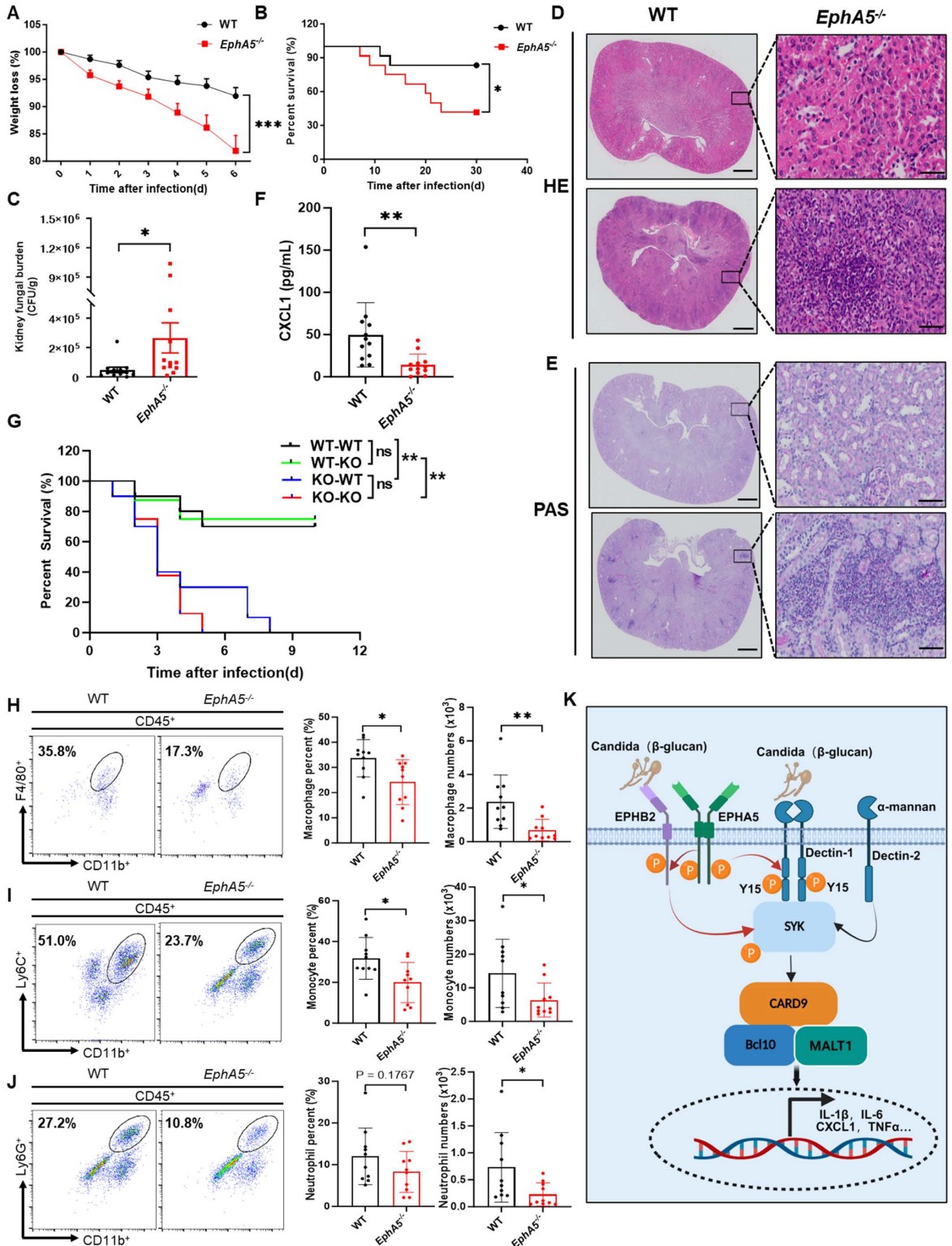

**Fig 7. *EphA5*-KO mice are susceptible to fungal sepsis.** (A) Weight loss (2 × 10⁵ CFU of *C. albicans* per mouse) and (B) survival (4 × 10⁵ CFU of *C. albicans* per mouse) of female WT (n = 12) and *EphA5*-KO (*EphA5⁻/⁻*) mice (n = 12) after infection with the live *C. albicans* SC5314 strain. (C) Statistics of kidney fungal burden in WT (n = 12) or *EphA5⁻/⁻* mice (n = 12) infected with 2 × 10⁵ CFU of *C. albicans* per mouse for 48 hours are shown. (D-G) WT and

*EphA5*-KO (*EphA5$^{-/-}$*) mice underwent the same treatment described in A and were sacrificed 48 hours after *C. albicans* infection. The representative images of the kidney sections were stained with hematoxylin and eosin (H&E) (D) or periodic acid-Schiff (PAS) (E) (n = 10). Scale bar = 500 μm or 20 μm for HE, and scale bar = 200 μm or 20 μm for PAS. (F) ELISA analysis of CXCL1 in the serum of WT and *EphA5$^{-/-}$* mice. (G) Survival of bone marrow (BM) chimeric mice following infection with 4 × 10$^5$ CFU of *C. albicans* per mouse is shown. Recipient mice were intravenously injected with 5 × 10$^6$ BM cells. *EphA5$^{+/+}$* mice were reconstituted with *EphA5$^{+/+}$* BM (WT→WT, n = 10), *EphA5$^{+/+}$* mice were reconstituted with *EphA5$^{-/-}$* BM (KO→WT, n = 10), *EphA5$^{-/-}$* mice were reconstituted with *EphA5$^{+/+}$* BM (WT→KO, n = 8), and *EphA5$^{-/-}$* BM was reconstituted with *EphA5$^{-/-}$* mice (KO→KO, n = 8). (H-J) WT or *EphA5$^{-/-}$* mice were injected intravenously with live *C. albicans* (2 × 10$^5$ CFU of *C. albicans* per mouse) and sacrificed 48 hours after infection. The cells from the kidneys were stained for flow cytometry analysis. A representative density plot, percentages and numbers of CD45$^+$CD11b$^+$F4/80$^+$ macrophages (H), CD45$^+$CD11b$^+$Ly6C$^+$ monocytes (I), and CD45$^+$CD11b$^+$Ly6G$^+$ neutrophils (J) are shown (n = 10). (K) A hypothetical model of the role of EPHA5 in the antifungal immune response is shown: EPHA5 plays a critical role in antifungal immunity by phosphorylating EPHB2 and Dectin-1 after fungal infection, which facilitates the recruitment and activation of Syk and subsequent activation of downstream antifungal signaling pathways. (K) was illustrated using images created with biorender software. *$P$ < 0.05; **$P$ < 0.01; ***$P$ < 0.001; ****$P$ < 0.0001 based on a two-tailed unpaired t test, two-way ANOVA and the log-rank (Mantel–Cox) test. All error bars indicate the S.E.M. of biological replicates. The data are representative of three independent experiments.

## Discussion

In the present study, we show that EPHA5 plays a critical role in the antifungal innate immune response by phosphorylating EPHB2 and Dectin-1 after fungal infection, which facilitates the recruitment and activation of Syk for downstream activation of antifungal signaling pathways (Fig 7K). In an early study, we showed that EPHB2 is a coreceptor of β-glucan and phosphorylates Syk to activate the downstream signaling pathway, but how EPHB2 is activated after fungal infection is unknown [16]. Here, we show evidence that, upon fungal infection, EPHA5 phosphorylates EPHB2 for its activation, which is critical for the activation of downstream signaling pathways (Fig 4). As EPHA5 cannot directly recognize β-glucan (Fig 4A), the mechanism of EPHA5 activation after fungal infection still needs to be addressed in the future. For example, we show that EPHA5 undergoes dimerization and autophosphorylation after fungal stimulation and then phosphorylates EPHB2 for its activation (Fig 4F–4J). However, the signal that triggers the dimerization and autophosphorylation of EPHA5 is still unknown. As EPHB2 and EPHA2 were reported to recognize fungal ligands, one hypothesis is that EPHA5 could also recognize fungal ligands other than β-glucan, which needs to be further investigated in the future [18,27].

In the present study, we show that EPHA5 phosphorylates EPHB2 and Dectin-1 after Dectin-1 ligand stimulation, which facilitates the recruitment and activation of Syk and subsequent activation of downstream antifungal signaling pathways. Surprisingly, we found that Dectin2 ligands, such as α-Mannan, is also required for the antifungal immune response triggered by Dectin-2/3 ligand stimulation (Figs 1B, 1D, 1G, 1I and 3C). Dectin-2/3 and FcγR, but not Dectin-1, are responsible for the α-Mannan-triggered antifungal immune response and signaling activation, and Dectin-2/3 does not contain an ITAM domain for the recruitment of Syk [13,14,32–34]. Therefore, the exact regulatory mechanism by which EPHA5 regulates Dectin-2/3 signaling is still unclear. We found that treatment with α-Manner promoted the autophosphorylation/dimerization of EPHA5 (Fig 4B and 4I) and that compared with WT EPHA5, the kinase-dead mutant of EPHA5 did not promote the activation of antifungal signaling (Fig 4D). These data indicate that EPHA5-mediated regulation of Dectin-2/3 signaling is also dependent on its kinase activity, as we show that EPHA5 interacts with Dectin-2 (S4D Fig), and one hypothesis is that EPHA5 also phosphorylates Dectin-2 at specific sites and facilitates the recruitment of downstream signaling molecules after Dectin-2/3 ligand stimulation; this hypothesis needs to be further investigated in the future.

Invasive fungal infections have become a significant global health threat, causing approximately 3.75 million deaths each year [2]. Owing to the limited number of antifungal drugs available in the clinic, clinical therapy for invasive fungal infections is extremely burdensome [1–3]. Immunotherapy has achieved great success in clinical anticancer treatment, and the same thread may also be applied in antifungal therapy. For example, in an earlier study, we showed that acetylation of α-Tubulin was required for antifungal immunity and that deacetylase-Sirt2 was responsible for the deacetylation of α-Tubulin and inactivation of the antifungal immune response. Therefore, small compounds of the Sirt2 inhibitors AGK2, AK-1, or AK-7 have good therapeutic effects on invasive fungal infections [14]. In another study, we showed that specific delivery of *RAC1* mRNA to monocytes or macrophages in vivo via LNPs provided significant protection against systemic

*C. albicans* infection [13]. As EPHB2 and EPHA5 are critical players in the antifungal immune response, one reasonable antifungal strategy may be screening activators of EPHB2 or EPHA5. Since EPHB2 and EPHA5 are both localized on the plasma membrane and have an extracellular domain, monoclonal antibodies might be good choices, as antibodies may crosslink proteins to form dimers or polymers, which leads to the activation of downstream pathways [30]. This hypothesis needs to be further explored, which may provide therapeutic methods for the treatment of ICIs.

## Materials and methods

### Ethics statement

All animal experiments were carried out following the general guidelines published by the Association for Assessment and Accreditation of Laboratory Animal Care. All of the mice were maintained under specific-pathogen-free conditions, and all animal studies were approved by approved by the Ethics Committee of Sichuan Provincial People's Hospital, University of Electronic Science and Technology of China.

### Mice

*EphA5*-knockout (*EphA5*-KO) mice were generated by Cyagen Biosciences, Inc., via CRISPR/Cas9 technology. All the mice were housed in a pathogen-free environment and cared for following institutional and national guidelines. Animal handling and experimental procedures were conducted in accordance with protocols approved by the Institutional Animal Care and Use Committee of Sichuan Provincial People's Hospital, University of Electronic Science and Technology of China. Male or female mice between 6–8 weeks of age were used for all experiments unless otherwise noted.

### Cell lines and primary cell cultures

THP-1, HEK293T, and RAW264.7 cells were cultured in RPMI-1640 or DMEM supplemented with 10% fetal bovine serum (FBS), 1% penicillin–streptomycin, and 2 mM L-glutamine. EPHA5 knockdown THP-1 cells were generated via CRISPR/Cas9, and GFP positive cells (EPHA5-CRISPR/Cas9 plasmids infected cells) were collected as a pool to perform the experiments. Wild-type (WT) and EPHA5 knockdown THP-1 cells were used to explore the role of EPHA5 in macrophage antifungal signaling. For the reconstitution experiments, EPHA5 knockdown (EPHA5-g2) THP-1 cells were transfected with lentiviral vectors encoding WT EPHA5 (EPHA5-WT) or kinase-inactivated EPHA5 (EPHA5-K708R). Bone marrow-derived macrophages (BMDMs) were isolated from WT and *EphA5*-KO mice as previously described. Briefly, bone marrow cells were harvested from the femurs and tibias of mice and cultured in RPMI-1640 with 10% FBS and 20 ng/mL M-CSF for 7 days to allow them to differentiate into macrophages. The cells were seeded in 6-well or 12-well plates 1 day prior to the experiment.

### Reagents

The following antibodies were used in this study: anti-HSP90, anti-Tyr, anti-Syk, anti-p-ERK, anti-ERK1/2, and anti-actin were purchased from Santa Cruz Biotechnology (cat. no. sc-13119, sc-2710S, sc-1240, sc-7383, sc-514302, and sc-58673, respectively). Flow cytometry antibodies, including anti-CD3 (clone 17A2), anti-CD45 (clone RM4–5), anti-CD11C (clone N418), anti-Ly6G (clone 1A8), anti-F4/80 (clone W20065B), anti-CD11b (clone M1/70), anti-CD19 (clone 6D5), anti-Gr-1 (clone RB6-8C5), anti-NK1.1-APC (clone PK136), anti-Ly6C (clone HK1.4), anti-Isotype (clone RTK2758), and anti-Dectin-1, were obtained from BioLegend (cat. no. 100222, 100510, 117318, 127614, 111604, 101235, 115507, 108424, 550627, 128036, 400507, 400512, and 355403). The Zombie Violet Fixable Viability Kit was purchased from BioLegend (cat. no. 423114). Curdlan and polyI:C were obtained from InvivoGen (cat. no. tlrl-cud and tlrl-picwlv), whereas α-Mannan and gentamicin were sourced from Sigma (cat. no. M3640 and E003632-1G). Recombinant mouse M-CSF proteins were purchased from PeproTech (cat. no. 315–02), and SYTOX Green was obtained from Thermo Scientific (cat. no. S7020).

The additional antibodies used included anti–p-IkB (Ser32), anti-p-SAPK/JNK (Thr183/Tyr185), anti-JNK1, anti-p-Syk, and anti-EPHB2, all of which were purchased from CST (cat. no. 5209S, 4668S, 3708S, 2710S, and 83029S). Anti-EPHA5, anti-Flag, and anti-HA antibodies were obtained from ABclonal (cat. no. A14238, AE024, and AE123). The anti–p-EPHB2 antibody was obtained from Invitrogen (cat. no. PA1–46222), and the anti-Dectin2 antibody was acquired from Abcam (cat. no. ab107572). Mouse CXCL1 and TNF-α ELISA kits were obtained from BioLegend (cat. no. 447507 and 430904). Finally, vancomycin and amphotericin B were purchased from MCE (cat. no. HY-17362 and HY-K1052).

## Culture and heat inactivation of *C. albicans* strain SC5314

The *C. albicans* strain SC5314 was cultivated on yeast-peptone-glucose agar (AGAR) plates, subsequently inoculated into yeast extract peptone dextrose (YPD) media, and incubated at 30°C overnight. Single colonies were subsequently selected and expanded in fresh YPD media. Upon reaching an optical density (OD) of 0.6–0.8, the yeast cells were harvested, subjected to three washing cycles with phosphate-buffered saline (PBS) to eliminate any residual media, and then resuspended in PBS buffer. The resuspended cells were subjected to heat treatment at 95°C for one hour, effectively inactivating the cells and producing heat-killed *Candida albican* (HKCA).

## RNA quantification and RT–qPCR

Total RNA was extracted from cells via TRIzol reagent (Invitrogen) according to the manufacturer's instructions. The RNA purity and concentration were assessed via a NanoDrop spectrophotometer. Reverse transcription of 1 μg of RNA was performed via SuperScript II Reverse Transcriptase (Thermo Fisher). Real-time quantitative PCR (qPCR) was conducted via SYBR Green Master Mix (Applied Biosystems) on a QuantStudio 6 Flex system (Thermo Fisher). Gene expression levels were normalized to those of the housekeeping genes Actb or GAPDH and calculated via the ΔΔCt method.

## ELISA

The levels of TNF-α and CXCL1 in mouse serum or culture supernatants were quantified via ELISA kits (BioLegend) following the manufacturer's protocols. The absorbance was measured at 450 nm via a microplate reader (Thermo Fisher). All samples were assayed in duplicate, and concentrations were determined on the basis of standard curves. In addition, ELISA was also performed to assess the binding of the recombinant EPHA5 and EPHB2 proteins to β-glucan and α-mannan.

## Immunoprecipitation and Western blot

The cells were lysed in ice-cold lysis buffer containing 0.5% Triton X-100, 20 mM HEPES (pH 7.4), 150 mM NaCl, 12.5 mM β-glycerophosphate, 1.5 mM MgCl₂, 10 mM NaF, 2 mM dithiothreitol (DTT), 1 mM sodium orthovanadate, 2 mM EGTA, and a cocktail of protease inhibitors (Roche, 11873580001) and phosphatase inhibitors (Roche, 4906845001). After centrifugation at 12,000 rpm for 15 min at 4°C, the supernatants were collected. For immunoprecipitation, the cell lysates were incubated with primary antibodies overnight at 4°C, followed by incubation with protein A/G Sepharose beads for 2 hours. The beads were then washed four times, and the bound proteins were eluted with 2 × sample buffer and analyzed by SDS–PAGE. For Western blotting, proteins were transferred onto PVDF membranes, which were blocked with 5% nonfat milk and incubated with primary antibodies overnight at 4°C. After incubation with HRP-conjugated secondary antibodies, signals were detected via enhanced chemiluminescence (ECL, GE Healthcare).

## Lentivirus-mediated gene knockdown in THP-1 cells

CRISPR/Cas9-mediated knockout of genes in THP-1 cells was performed via the pLenti-CRISPR-GFP vector. Lentiviral particles were generated by transfecting HEK293T cells with packaging plasmids and the pLenti-CRISPR-GFP plasmid.

Viral supernatants were collected at 48 or 72 hours post transfection, filtered, and used to infect THP-1 cells. Transduced cells were sorted for GFP expression via flow cytometry, and cells were isolated in 96-well plates. Successful knockout cells were confirmed by Western blot and Sanger sequencing.

### Human PBMC study

Human PBMCs were isolated from blood samples of healthy donors via Ficoll–Paque PLUS (GE Healthcare). The cells were washed twice in PBS and resuspended in RPMI 1640 supplemented with 10% FBS. PBMCs were transfected with siRNA via RFect SP Transfection Reagent (BioGenerator Biotechnology) according to the manufacturer's protocol. Three days post transfection, the PBMCs were stimulated with Curdlan (100 µg/mL), HKCA (E; T=2), or α-Mannan (100 µg/mL) for gene expression analysis. The study was approved by the Ethics Committee of Sichuan Provincial People's Hospital, University of Electronic Science and Technology of China.

### In vitro kinase assay and in vitro binding assay

To investigate the kinase activity and binding interactions of EPHA5 and EPHB2, we performed in vitro kinase and binding assays. For protein purification, BL21 competent *E. coli* cells were transformed with recombinant pSmartI plasmids containing His-EPHB2. The cells were cultured overnight at 37°C in LB media and then subcultured until the optical density at 600 nm (OD600) reached 0.6–0.8. Isopropyl β-D-thiogalactoside (IPTG) was added, and the culture was shaken at 16°C overnight. After centrifugation and lysis, His-tagged proteins were purified via Ni-NTA agarose (QIAGEN) following the manufacturer's protocols. Protein concentrations were determined via a BCA protein assay kit (Thermo Fisher).

To investigate the kinase activity of EPHA5-WT and EPHA5-K708R in response to different stimuli, EPHA5 knockdown (EPHA5-g2) THP-1 cells expressing EPHA5-WT or EPHA5-K708R were differentiated with PMA for 48 hours, and stimulated with HKCA or α-Mannan. Following cell lysis on ice, the lysates were incubated with Flag beads overnight and washed twice with kinase buffer. Kinase reaction buffer and EPHB2-His proteins were added, and the mixture was incubated at 30°C for 1 hour before the reaction was terminated by heating at 90°C for 5 minutes. SDS sample buffer was then added, and Western blot analysis was performed.

### Mouse model of systemic *C. albicans* infection and histopathology

The *C. albicans* SC5314 strain was cultured overnight at 30°C in 25 ml of YPD broth, counted via a hemocytometer, and diluted to $2 \times 10^6$ cells/ml. The samples were washed twice with PBS and resuspended in $1 \times$ PBS. The mice were injected with 0.1 ml of the fungal cell suspension via the lateral tail vein and monitored for signs of illness, weight changes, and survival over 30 days. For fungal burden assessment, the mice were sacrificed 48 hours post infection. Kidneys were harvested, homogenized, serially diluted in water, and plated on YPD agar to determine colony-forming units (CFUs) per gram of tissue after incubation at 30°C for 2 days. Immunohistochemical analysis of the kidneys was conducted by ServiceBio (China). Kidneys were fixed in 4% paraformaldehyde, embedded in paraffin, sectioned, and stained with H&E and periodic acid-Schiff (PAS). Slides were scanned via a Pannoramic MIDI (3D HISTECH).

### Bone marrow transplantation

Recipient mice (WT mice) were administered sterile water containing vancomycin (0.5 mg/mL), gentamicin (170 µg/mL), or amphotericin B (10 µg/mL) for 1 week prior to transplantation. They were then subjected to lethal irradiation (8 Gy, Raycision Medical Technology Co., Ltd., SHARP 100) to eliminate endogenous bone marrow stem cells and received two doses 4 hours apart. Four hours after the second irradiation, $5 \times 10^6$ bone marrow cells from the donor mice (WT and *EphA5*⁻ᐟ⁻ mice) were injected intravenously into the tail vein of the recipients. Following transplantation, the mice received

antibiotic-treated water for 3 weeks to prevent infection. After 8 weeks, the mice were challenged with an intravenous injection of $4 \times 10^5$ *C. albicans* SC5314 cells, and their survival rates were monitored.

## Flow cytometry

Two days postinfection, the mice were sacrificed and perfused with $1 \times$ PBS. Kidneys were homogenized via ice-cold tissue grinders, and the resulting cell suspension was filtered through a 70 µm cell strainer. The cells were collected via centrifugation at $400 \times g$ for 5 minutes at 4°C. The samples were then digested with a digestion solution (PBS with 2% FBS, 0.05 mg/ml collagenase, 1 mg/mL dispersion, and 20 U/mL DNase) on a shaker at 37°C and 100 rpm for 20 minutes. Lymphocytes were isolated via 40% and 80% Percoll gradients, followed by surface staining for 30 minutes at 20°C. To exclude dead cells, Zombie Violet Fixable Viability Reagent (1:1000; BioLegend) was added. Flow cytometry data were analyzed via Novoexpress software.

## Statistical analysis

All experiments were performed with at least two independent biological replicates. The data were analyzed via GraphPad Prism software, and statistical significance was assessed via two-tailed unpaired t tests, two-way ANOVA, or log-rank (Mantel–Cox) tests, as appropriate. P values < 0.05 were considered statistically significant.

## Supporting information

**S1 Fig. EPHA5 is essential for activating antifungal signaling in macrophages.** (A-B) Quantification of phosphorylated protein levels (P-IκBα, P-ERK, P-JNK, and P-Syk) on the basis of grayscale intensity analysis from Fig 1A and 1B was shown. (C-E) Quantification of phosphorylated proteins (P-IκBα, P-JNK, and P-ERK) from the grayscale intensity values in Fig 1C–1E was shown. (F-G) Western blot analysis of bone marrow-derived macrophages (BMDMs) from WT and *EphA5*-KO mice stimulated with LPS (20 ng/mL, F) or PolyI:C (100 ng/mL, G) for the indicated time points was shown to assess the activation of key signaling proteins. Data are presented as the mean ± standard error of the mean (S.E.M.) from biological replicates. Statistical significance was determined by a two-tailed unpaired t test (*P < 0.05; **P < 0.01; ***P < 0.001; ****P < 0.0001).
(TIF)

**S2 Fig. EPHA5 is required for fungal stimulation-induced cytokine and chemokine production in mouse peritoneal macrophages.** (A-C) RT–qPCR analysis of CXCL1, TNF-α, IL-1β, and IL-6 expression in peritoneal macrophages from WT and *EphA5*-KO mice stimulated with Curdlan (100 µg/mL) (A), α-Mannan (100 µg/mL) (B), or HKCA (C) at the indicated time points was shown. (D-E) ELISA quantification of CXCL1 (D) and TNF-α (E) levels in supernatants from peritoneal macrophages stimulated with Curdlan (100 µg/mL), HKCA, or α-Mannan (100 µg/mL) for 24 hours was shown. The data are shown as the means ± S.E.M.s from biological replicates. Statistical significance was determined by a two-tailed unpaired t test (*P < 0.05; **P < 0.01; ***P < 0.001; ****P < 0.0001). The results are representative of three independent experiments.
(TIF)

**S3 Fig. EPHA5 regulates antifungal immune responses through its kinase activity.** (A-B) Quantitative analysis of the grayscale intensity ratio of P-Y to IP-Flag in Fig 4A and 4B was shown. (C) Quantification of phosphorylated proteins (P-IκBα, P-ERK, and P-JNK) via grayscale intensity values from Fig 4C was shown.
(TIF)

**S4 Fig. Dectin-1/Dectin-2 expression in EPHA5 knockdown THP-1 cells and the interaction between Dectin-2 and EPHA5.** (A) Flow cytometry analysis of Dectin-1 and Dectin-2 expression in control (Vector) or EPHA5 knockdown THP-1

cells (EPHA5-g2 and EPHA5-g3) was shown. (B) Western blot validation of the EPHA5 knockout efficiency in THP-1 cells. (C) HEK293T cells were co-transfected with Dectin2-Flag and EPHA5-HA, followed by immunoprecipitation with an anti-HA antibody, and western blot analysis was shown for the indicated proteins.
(TIF)

**S5 Fig. Immune cell distribution in EPHA5-deficient mice and bone marrow chimeric experiments.** (A-C) Flow cytometric analysis of immune cell subsets in the serum, spleen and lymph nodes (LNs) of WT and $EphA5^{-/-}$ mice was shown. The percentages of immune cells, including T cells, B cells, NK cells, dendritic cells (DCs), myeloid-derived suppressor cells (MDSCs), macrophages, granulocytes, and monocytes, were shown. Each dot represents an individual mouse (n = 4 per group). (E-H) Flow cytometric analysis of immune cell subsets in bone marrow chimeric mice following irradiation and reconstitution with WT or $EphA5^{-/-}$ bone marrow was shown. The immune cell distribution in the serum of the recipient mice was assessed. Each dot represents an individual mouse (n = 5 per group). Statistical significance was determined by a two-tailed unpaired t test (ns, not significant; *$P < 0.05$; **$P < 0.01$; ***$P < 0.001$; ****$P < 0.0001$).
(TIF)

**S6 Fig. Flow cytometry gating strategy for macrophages, granulocytes, and monocytes.** (A) The gating strategy for macrophages was shown: single cells were first gated on FSC-A and SSC-A, followed by the exclusion of dead cells via Zombie Violet staining. The immune cells were identified as CD45$^+$, and the macrophages were further gated as F4/80$^+$CD11b$^+$ cells. (B) The gating strategy for granulocytes and monocytes was shown: single cells were gated on FSC-A and SSC-A, with live cells identified by Zombie Violet staining. The immune cells were gated as CD45$^+$, with granulocytes defined as Ly6G$^+$CD11b$^+$ and monocytes as Ly6C$^+$CD11b$^+$.
(TIF)

**S1 Data. Raw values for plots displayed in this manuscript.**
(XLSX)

## Author contributions

**Conceptualization:** Ru Gao, Ming Yi, Chenhui Wang.

**Data curation:** Ru Gao, Heping Wang, Zhihui Cui, Ming Yi.

**Formal analysis:** Ru Gao, Yanyun Du, Ruirui He, Lingyun Feng, Bo Zeng, Ming Yi.

**Funding acquisition:** Yanyun Du, Ruirui He, Lingyun Feng, Bo Zeng, Yangyang Li, Ming Yi, Chenhui Wang.

**Investigation:** Ru Gao, Yangyang Li, Guoling Huang, Ting Pan, Yuan Wang, Ming Yi.

**Methodology:** Ru Gao, Bo Zeng, Yangyang Li, Ming Yi.

**Project administration:** Ru Gao, Ming Yi.

**Resources:** Chenhui Wang.

**Software:** Ru Gao, Bo Zeng, Ming Yi.

**Supervision:** Ming Yi, Chenhui Wang.

**Validation:** Ru Gao, Bo Zeng, Ming Yi.

**Visualization:** Ru Gao, Ming Yi.

**Writing – original draft:** Ru Gao, Ming Yi, Chenhui Wang.

**Writing – review & editing:** Ming Yi, Chenhui Wang.

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
