## [Decision Letter · Decision Letter 0]

EPHA5 regulates antifungal innate immunity by phosphorylating EPHB2 and Dectin-1

Dear Dr. Wang,

Thank you for submitting your manuscript to PLOS Pathogens. After careful consideration, we feel that it has merit but does not fully meet PLOS Pathogens's publication criteria as it currently stands. Therefore, we invite you to submit a revised version of the manuscript that addresses the points raised during the review process.

Please submit your revised manuscript within 60 days Apr 08 2025 11:59PM. If you will need more time than this to complete your revisions, please reply to this message or contact the journal office at plospathogens@plos.org. Please include the following items when submitting your revised manuscript:

We look forward to receiving your revised manuscript.

Kind regards,

James B. Konopka

Academic Editor

PLOS Pathogens

Michal Olszewski

Section Editor

PLOS Pathogens

Editor-in-Chief

PLOS Pathogens

orcid.org/0000-0003-2946-9497

Editor-in-Chief

PLOS Pathogens

orcid.org/0000-0002-7699-2064

**Additional Editor Comments:**

The three reviewers agreed that your manuscript addresses a significant topic, but each had significant concerns about the presentation of the data and the use of appropriate controls. For example, Reviewer 1 had concerns about the bone marrow chimeric mice experiments. Reviewer 2 had concerns about the flow cytometry experiments. Reviewer 3 stated it was important to provide quantitative analysis of the Western blots and an additional control stimulant that does not engage Dectin1 or Dectin 2. There were also minor concerns that should be addressed.

In addition, the reviewers indicated that the Discussion section was very inadequate. It shoul be revised to summarize the key conclusions of the manuscript and put the results in context with previously published research. This will require adding new information to the Discussion to address these concerns.

**Journal Requirements:**

At this stage, the following Authors/Authors require contributions: Ru Gao, Heping Wang, Zhihui Cui, Yanyun Du, Ruirui He, Lingyun Feng, Bo Zeng, Yangyang Li, Guoling Huang, Ting Pan, Yuan Wang, Ming Yi, and Chenhui Wang. Please ensure that the full contributions of each author are acknowledged in the "Add/Edit/Remove Authors" section of our submission form.

https://journals.plos.org/plospathogens/s/submission-guidelines#loc-parts-of-a-submission

5) We have noticed that you have uploaded Supporting Information files, but you have not included a list of legends. Please add a full list of legends for your Supporting Information files after the references list.

6) In the online submission form, you indicated that "The datasets generated during and/or analyzed during the current study are available from the corresponding author upon reasonable request". All PLOS journals now require all data underlying the findings described in their manuscript to be freely available to other researchers, either

- In a public repository

- Within the manuscript itself

- Uploaded as supplementary information.

7) Please ensure that the funders and grant numbers match between the Financial Disclosure field and the Funding Information tab in your submission form. Note that the funders must be provided in the same order in both places as well. State what role the funders took in the study. If the funders had no role in your study, please state: "The funders had no role in study design, data collection and analysis, decision to publish, or preparation of the manuscript.".

**Reviewers' Comments:**

Reviewer's Responses to Questions

**Part I - Summary**

Reviewer #1: The manuscript “EPHA5 regulates antifungal innate immunity by phosphorylating EPHB2 and Dectin-1” investigated the function of EphA5 during antifungal immunity. The authors found that EPHA5 plays a critical role in antifungal immunity by phosphorylating EPHB2 and Dectin-1 after fungal infection. The authors used overexpression of WT and kinase dead receptor variants in different cell line to show physical interaction and receptor crosstalk of EphA5, EphB2, and dectin-1 upon fungal stimulation. Finally, the authors used Epha5 KO mice, as well as chimeric mice with deficiency in the hematopoietic compartment, to show that EphA5 is required for protection against lethal candidiasis. The manuscript is well written. However, some questions remain.

Reviewer #2: The manuscript by Gao et al describes the novel activation of EPHB2 and Dectin-1 by EPHA5 and the important functional consequences of this activation during fungal immunity. The authors provide strong evidence for EPHA5 phosphorylating dectin-1 at tyr 15 which promotes the efficient recruitment of Syk and downstream signalling. The authors further strengthen their study using EPHA5 deficient in vivo models. Overall, the results are very interesting and provide insight into a novel mechanism of CLR biology that has important consequences. However, the authors suggest EPHA5 has effect beyond Dectin-1 but do not investigate this. Please find my comments for the authors below.

Reviewer #3: The manuscript by Gao et al. demonstrates the mechanistic role of EPHA-5 in host defense against C. albicans. Coupled with in vivo animal data, immunological data, and biochemistry, this group defines a non-redundant role for EPHA-5 to serve as a key protein to mediate phosphorylation of EPHB2 and Dectin-1. While there is plenty of data packed into this manuscript, there needs some additional attention to important controls and additional quantitation of biochemical data. These additional data would greatly strengthen the rigor of the stated conclusions.

**Part II – Major Issues: Key Experiments Required for Acceptance**

Reviewer #1: Fig1: It is a bit surprising that in their Epha5 KO CRISPR THP1 cell lines a slight band within an immunoblot remains (same for the BMDMs). This needs clarification or probing with a different antibody. Also, the authors should check if EphA5 deletion reduces cell surface expression of CLRs including dectin-1/2/3.

-Fig6: Unclear why data from weight loss and survival are shown from different Candida inoculum. Please add inoculum for kidney fungal burden in the legend not only in the text. Please provide total numbers of infiltrating neutrophils, monocytes, and macrophages rather than only percentages.

-For the bone marrow chimeric mice experiments: the authors need to provide data of archived chimerism. Also, missing groups are KO->KO and WT->KO. Is it possible that EphA5 has a function on radio-resistant cells during antifungal immunity as well? As seen for EphA2 (PMID: 36138043)

Reviewer #2: 1) The authors use curdlan (Dectin-1), a-mannan (Dectin-2) and heat killed candida (likely multiple CLR ligands) throughout the manuscript. The authors show that EPHA5 mediates signalling in response to all three CLR ligands. Like most CLRs, Dectin-2 does not contain an itam domain and does not require tyr15 phosphorylation but instead associates with FcRg for signalling. The authors should clarify why they used a-mannan and discuss the implications of EPHA5 regulating broader CLR signalling.

2) The discussion requires extensive development to appropriately place the findings of this study within our current understanding of CLR signalling and anti-fungal immunity.

Reviewer #3: 1. Introduction Line 10 – I understand that the authors are trying to indicate that antimicrobial resistance is the driving force. I would hesitate to use the word “misuse” as it implies intent. I don’t think that is what the authors intend.

2. Results Line 11- Do you mean Fig 1C-E instead of Fig1A-B.

3. Please add catalog numbers to all antibodies and key reagents.

4. Which protease inhibitors were added for IP/Western experiments?

5. E;T should be used in place of MOI when heat-killed organisms are used. As they are dead, there is no infection.

6. Adjust spacing to Fig 1A, B, F, G, and H to group the whole protein western blot with the phosphorylated protein interrogated.

7. What does g2, g3 mean in figure 1?

8. Quantitation of blots for all would be helpful, especially, Figures 1F, 1G, 1H, 3D, 3H.

9. Is Figure 4 done with g1 or g2 THP-1 KO cells?

**Part III – Minor Issues: Editorial and Data Presentation Modifications**

Reviewer #1: -The Candida glabrata/ auris data feel misplaced this late in the manuscript and should be integrated in earlier figures/parts.

Reviewer #2: 1) The flow cytometry in figure 6H and 6I is not clear.

a) The authors should add their gating strategies as a supplementary figure.

b) I am unconvinced by the f4/80 staining, the authors should include f4/80 isotype staining to better gate on

f4/80+ cells.

c) Figure 6I likely contains neutrophils and monocytes, did the authors use Ly6G to remove neutrophils? This

should be shown.

2) The methods state fungal burden was assessed after 5 days but the manuscript says 48 hours.

Reviewer #3: 1. The use of fungal ligands demonstrates differences between WT and KO cells. However, since EPHA5 has not been previously implicated in immune responses, it would be prudent to use a stimulant that does not engage Dectin1 or Dectin 2 (e.g. LPS - engaging TLR4). The expectation is that this activation would still be intact in KO cells and comparable to WT cells. This would increase the rigor of these conclusions.

2. Some of the differences in biochemistry are subtle. Quantification of these blots would go a long way to establish significance, as stated by the authors. Significance indicates a mathematical difference, but these data are not provided.

3. Since the KO mice have not been examined for any immunological defects, can the authors either cite a paper or provide data that these mice do not have any gross immunological defects that could skew the interpretation (i.e. number of circulating neutrophils, changes in the spleen, lymph nodes, etc.).

4. How do the authors reconcile the TNF data shown in Figure 3A with S1C,E?

5. The discussion misses the opportunity to discuss these findings in the broader context and instead focuses on potential use in therapeutics. I would love to hear their thoughts on possible ligands, for example.

PLOS authors have the option to publish the peer review history of their article (what does this mean? ). If published, this will include your full peer review and any attached files.

**Do you want your identity to be public for this peer review?** For information about this choice, including consent withdrawal, please see our Privacy Policy .

Reviewer #1: No

Reviewer #2: No

Reviewer #3: **Yes: ** Jatin Mahesh Vyas

**Figure resubmission:**

**Reproducibility:**



---

## [Decision Letter · Decision Letter 1]

Dear Professor Wang,

We are pleased to inform you that your manuscript 'EPHA5 regulates antifungal innate immunity by phosphorylating EPHB2 and Dectin-1' has been provisionally accepted for publication in PLOS Pathogens.

Best regards,

James B. Konopka

Academic Editor

PLOS Pathogens

Michal Olszewski

Section Editor

PLOS Pathogens

Sumita Bhaduri-McIntosh

Editor-in-Chief

PLOS Pathogens

orcid.org/0000-0003-2946-9497

Michael Malim

Editor-in-Chief

PLOS Pathogens

orcid.org/0000-0002-7699-2064

Reviewer Comments (if any, and for reference):

Reviewer's Responses to Questions

**Part I - Summary**

Reviewer #1: The authors addresses this reviewers queries.

Reviewer #2: (No Response)

Reviewer #3: The authors have addressed my concerns. No further issues need to be resolved prior to publication.

**Part II – Major Issues: Key Experiments Required for Acceptance**

Reviewer #1: (No Response)

Reviewer #2: (No Response)

Reviewer #3: The authors have addressed my concerns. No further issues need to be resolved prior to publication.

**Part III – Minor Issues: Editorial and Data Presentation Modifications**

Reviewer #1: (No Response)

Reviewer #2: (No Response)

Reviewer #3: The authors have addressed my concerns. No further issues need to be resolved prior to publication.

PLOS authors have the option to publish the peer review history of their article (what does this mean? ). If published, this will include your full peer review and any attached files.

**Do you want your identity to be public for this peer review?** For information about this choice, including consent withdrawal, please see our Privacy Policy .

Reviewer #1: No

Reviewer #2: No

Reviewer #3: **Yes: ** Jatin Mahesh Vyas, MD, PhD

---

## [Editor Report · Acceptance letter]

Dear Professor Wang,

We are delighted to inform you that your manuscript, "EPHA5 regulates antifungal innate immunity by phosphorylating EPHB2 and Dectin-1," has been formally accepted for publication in PLOS Pathogens.

Best regards,

Sumita Bhaduri-McIntosh

Editor-in-Chief

PLOS Pathogens

orcid.org/0000-0003-2946-9497

Michael Malim

Editor-in-Chief

PLOS Pathogens

orcid.org/0000-0002-7699-2064